# OpenReview forum: "Return Augmented Decision Transformer for Off-Dynamics Reinforcement Learning"
_TMLR — Accepted by TMLR_

### Review · Reviewer_7hkf · 2025-11-19

**Summary Of Contributions:**

The paper proposes Return Augmented Decision Transformer (REAG) for offline off-dynamics RL, where a large source dataset with shifted dynamics supplements limited target data. Prior reward-augmentation methods (DARC, DARA) designed for dynamic programming–based RL cannot directly transfer to return-conditioned supervised learning (RCSL) methods like Decision Transformer, since RCSL policies explicitly condition on return-to-go. REAG augments returns rather than rewards by transforming source returns to align with target distribution.

Two implementations are proposed: REAG*_Dara adapts DARA's classifier approach (ψ(g) = g + η∑Δr), while REAG*_MV matches return distributions via Gaussian approximation. Theorem 4.5 shows suboptimality improves from O(1/√N_T) to O(1/√N_S) when N_S >> N_T. Experiments on D4RL MuJoCo (Walker2D, Hopper, HalfCheetah) with BodyMass/JointNoise shifts show both variants improve DT, Reinformer, and QT, with REAG*_MV generally outperforming REAG*_Dara.

**Key Strengths:**
- Clear motivation adapting augmentation techniques to RCSL policy classes
- Principled derivations with finite-sample guarantees matching non-shifted regime
- Comprehensive ablations on shift magnitude, clipping, consistency, value function quality

**Key Weaknesses:**
- Incremental novelty—REAG*_Dara directly translates DARA; Gaussian approximation lacks justification
- Missing recent off-dynamics baselines (H2O, VGDF, PAR, BOSA); comparisons only to 2019-2020 methods
- Inconsistent improvements (~30% degradation in Table 2) with no significance testing
- Narrow scope (MuJoCo locomotion only, two simple shifts, ad-hoc clipping bounds)

**Additional Comments:**

Overall, I think this is work is useful: extends DT to off-dynamics settings. Adding modern baselines and making experiments more thorough would make this a solid paper for TMLR.

**Audience:**

Yes

**Audience Explanation:**

The paper tackles a real gap: applying Decision Transformer to off-dynamics RL. This matters for robotics sim-to-real transfer and any domain where target interaction is expensive. The insight that reward-augmentation techniques need adaptation for RCSL is useful and helps practitioners understand these methods. The finding that return-level augmentation can beat reward-level approaches for DT is worth knowing, as is showing these methods can compete with traditional offline RL (Figure 1).

That said, the contribution is incremental, in that it adapts existing ideas rather than new insights. The scope is narrow (just MuJoCo locomotion). That said, it's a concrete contribution that extends DT methods to an important setting.

**Broader Impact Concerns:**

No Broader Impact Statement needed. The main risk is that better sim-to-real transfer might lead to deploying insufficiently validated policies in safety-critical applications, but this applies to all RL methods. No major ethical concerns.

**Claims And Evidence:**

No

**Claims Explanation:**

The theoretical claims are sound but straightforward applications of existing results. The Gaussian approximation (Eq 4.4) has no validation of when it's appropriate. The experiments have some gaps:

**Missing state-of-the-art baselines**: Comparisons are against 2019-2020 methods (BEAR, AWR, BCQ, CQL, MOPO) augmented with DARA. Recent off-dynamics methods (H2O, VGDF, PAR, BOSA, IGDF, SRPO) mentioned in related work are absent. Without these, we can't tell if REAG is actually competitive with current approaches.

**Inadequate statistical rigor**: Table 2 shows high variance with ~30% of results degrading (marked ↓). Example: Walker2D M-R JointNoise REAG_MV^Reinf gets 47.591±10.244 versus baseline 54.801±3.217. The paper claims "consistent enhancement" but the evidence shows otherwise. No significance tests despite variance often exceeding mean differences.

**Limited scope**: Only MuJoCo locomotion with two simple shifts (mass scaling, action noise). The claims need validation beyond this narrow setting.

**Implementation underspecified**: Clipping bounds (Table 6) look ad-hoc with no principled method. Number of rollouts for σ estimation not reported. No wall-clock time analysis for CQL training overhead.

The ablations (Section 5.4) are solid, and value function validation (Figure 6) is convincing. But missing modern baselines and lack of significance testing are serious problems.

**Requested Changes:**

**Major**

1. **Add recent off-dynamics baselines**: Include 2-3 recent methods (H2O, VGDF, PAR, or BOSA) in main experiments to show how REAG compares to ~state-of-the-art.

2. **Statistical significance testing**: Report p-values or confidence intervals for Table 2.

3. **Fix overclaims**: Abstract and Section 5.3 claim "consistent enhancement" but ~30% of Table 2 shows degradation.

**Misc**

4. **Justify Gaussian approximation**: Show histogram plots of return distributions or provide theoretical analysis of when this approximation works.

5. **Expand experimental scope**: Add one non-locomotion task and one complex dynamics shift (morphology, contact parameters) to show this generalizes.

6. **Failure mode analysis**: Explain when REAG degrades performance (e.g., Reinformer in Walker2D M-R) so users know when to apply it.

---

> ### Author Response · Authors · 2026-01-06
> **Response to Reviewer 7hkf (1/3)**
>
> > **Q1: Incremental Novelty — $\text{REAG}^{*}_{\mathrm{Dara}}$ Directly Translates DARA**
>
> **Response.**
> REAG introduces a **return-level augmentation** framework tailored to **DT-type, return-conditioned policies**, which is fundamentally different from the **reward-level augmentation** used in DARA. The variant $\text{REAG}^{*}_{\mathrm{Dara}}$ is included as a *controlled adaptation* of reward augmentation to the DT setting for comparison purposes, rather than as the primary contribution.
>
> Our main contribution is $\text{REAG}^{*}_{\mathrm{MV}}$, which performs **return distribution matching** and is specifically designed for the **RCSL** framework. Empirically, the results in **Figure 1** show that directly applying reward-level augmentation is insufficient for DT-type methods, whereas return-level augmentation is necessary for effective off-dynamics RCSL.
>
> > **Q2: Comparisons to Recent Off-Dynamics Methods**
> > Comparisons are against 2019–2020 methods (BEAR, AWR, BCQ, CQL, MOPO) augmented with DARA. Recent off-dynamics methods (H2O, VGDF, PAR, BOSA, IGDF, SRPO) mentioned in related work are absent. Without these, it is unclear whether REAG is competitive with current approaches.
>
> **Response.**
> Thank you for the comment. In response, we conducted additional experiments comparing **REAG** with more recent off-dynamics methods, including **H2O** and **IGDF**. The corresponding results are reported in **Table 9** of the revised manuscript. In addition, **Figure 1** shows that across the evaluated settings, REAG remains competitive and, in many cases, achieves stronger performance in addressing off-dynamics reinforcement learning challenges.
>
> > **Q3: High Variance and Claims of Consistent Enhancement**
> > Table 2 shows high variance with ~30% of results degrading (marked ↓). Example: Walker2D M-R JointNoise, $\text{REAG}^{\text{Reinf}}_{\mathrm{MV}}$ achieves $47.591 \pm 10.244$ versus the baseline $54.801 \pm 3.217$. The paper claims “consistent enhancement,” but the evidence appears mixed. In addition, no significance tests are reported, despite variance often exceeding mean differences.
>
> **Response.**
> Thank you for the detailed comment. To address this concern, we emphasize that our claims of improvement are based on **overall performance trends** rather than per-setting mean comparisons alone.
>
> Specifically, we provide an aggregated comparison in **Figure 1**, which reports **average normalized rank scores** for all evaluated algorithms across the **Medium**, **Medium-Replay**, and **Medium-Expert** datasets under both **BodyMass** and **JointNoise** shift settings in the **Walker2D**, **Hopper**, and **HalfCheetah** environments. Within each setting, algorithms are ranked based on performance, with the best-performing method assigned rank 1. Tied scores receive the same rank, and the remaining ranks are adjusted accordingly. The original ranks (from 22 algorithms) are then normalized to a scale of 1 to 22, and the figure reports the average normalized rank across all environments and settings. Lower rank scores indicate better overall performance.
>
> Compared to raw mean–variance comparisons, this **rank-based metric** provides a more general and robust evaluation that is less sensitive to high variance in individual tasks. As shown in Figure 1, our proposed methods consistently achieve stronger average rank scores than the baselines in the overall evaluation.
>
> To avoid overstatement, we have revised the manuscript to clarify that $\text{REAG}_{\mathrm{MV}}$ delivers **more stable and robust improvements in most cases**, rather than claiming uniform gains across all individual settings.

---

> > ### Author Response · Authors · 2026-01-06
> > **Response to Reviewer 7hkf (2/3)**
> >
> > > **Q4: Scope of Evaluation and Generalization Beyond Simple Shifts**
> > > Only MuJoCo locomotion with two simple shifts (mass scaling, action noise) are evaluated. The claims require validation beyond this narrow setting. In particular, adding at least one non-locomotion task and one complex dynamics shift (e.g., morphology or contact-parameter changes) would help demonstrate generality.
> >
> > **Response.**
> > Thank you for the comment. For **non-locomotion tasks**, to our knowledge, there are currently no established benchmarks that explicitly study **offline off-dynamics reinforcement learning** with well-defined source–target dynamics shifts. As a result, existing work in this area has largely relied on **MuJoCo locomotion environments** to enable controlled and reproducible evaluation.
> >
> > To assess robustness beyond simple mass scaling and action noise, we additionally evaluate **more complex dynamics shifts**. Specifically, we conduct ablation studies on **HalfCheetah** under a **Kinematic Shift**, where the rotation range of the back thigh joint is modified from $[-0.52,\,1.05]$ to $[-0.0052,\,0.0105]$. This modification induces a substantial morphology-related dynamics change.
> >
> > The corresponding results are reported in **Table 11** of the revised manuscript and show that **REAG remains effective under more complex dynamics shifts**, providing additional evidence of its robustness beyond the standard off-dynamics settings.
> >
> > > **Q5: Clipping Bounds, Rollouts, and Computational Cost**
> > > Clipping bounds (Table 6) appear ad hoc without a principled derivation. The number of rollouts used for $\sigma$ estimation is not reported. In addition, no wall-clock time analysis is provided for the CQL training overhead.
> >
> > **Response.**
> > Thank you for the comment. We clarify these implementation details as follows.
> >
> > - **Number of rollouts for $\sigma$ estimation.**
> >   We use **1,000 rollouts** to estimate $\sigma$ across all experiments.
> >
> > - **CQL training cost.**
> > The CQL training times for each setting are summarized in the revised manuscript. All CQL models are trained under a **uniform hardware configuration** consisting of an **NVIDIA RTX A5000 GPU (24 GB memory)** and an **AMD Ryzen Threadripper 3960X CPU (24 cores)**. Under this setup, training a CQL model on the **target dataset** requires approximately **2.48 GPU-hours**, while training on the **source dataset** requires approximately **9.39 GPU-hours**.
> >
> > These computational details, including rollout counts and wall-clock training costs, have been added to **Section C.4** of the revised manuscript.

---

> > > ### Author Response · Authors · 2026-01-06
> > > **Response to Reviewer 7hkf (3/3)**
> > >
> > > > **Q6: Statistical Significance Testing**
> > > > Report p-values or confidence intervals for Table 2.
> > >
> > > **Response.**
> > > Thank you for the comment. We have revised the experimental results to explicitly report **95% confidence intervals (CI)** for all entries in **Table 7 (Appendix D)**, computed over **five random seeds**. The reported CIs quantify the uncertainty of the mean performance and enable assessment of the reliability of the observed improvements.
> > >
> > > We observe that, in most settings, our proposed methods improve over the corresponding DT-type baselines, with improvements that exceed the CI radius. In several cases—particularly under the **Medium-Replay** and **Medium-Expert** settings on **Walker2D**, **Hopper**, and **HalfCheetah**—the confidence intervals exhibit limited overlap with those of the baselines, further supporting the statistical reliability of the gains.
> > >
> > > > **Q7: Justification of the Gaussian Approximation**
> > > > Justify the Gaussian approximation by showing histogram plots of return distributions or providing theoretical analysis of when this approximation is valid.
> > >
> > > **Response.**
> > > We adopt a Gaussian assumption based on empirical evidence indicating that the return distribution is approximately normal. Specifically, we analyze the distribution of **trajectory-level cumulative rewards** using the **D4RL** dataset, where the *medium* datasets are collected by a medium-performance **SAC** policy.
> > >
> > > For the **Hopper** and **HalfCheetah** environments, we compute final cumulative rewards for each trajectory and visualize their distributions using **histograms**. The resulting distributions are largely **unimodal** and approximately **symmetric**, suggesting that a Gaussian distribution provides a reasonable approximation in these settings.
> > >
> > > This empirical analysis has been included in **Figure 8** in **Section D.3** of the revised manuscript.
> > >
> > > > **Q8: Failure Mode Analysis**
> > > > Explain when REAG degrades performance (e.g., Reinformer on Walker2D Medium-Replay), so users know when to apply it.
> > >
> > > **Response.**
> > > REAG is most effective when the **source and target domains exhibit sufficient state(-action) occupancy overlap**. This requirement is formalized in our theoretical analysis through the **domain occupancy overlap coefficient** (Assumption 4.2 and Remark 4.6), which characterizes when source data can provide reliable information for the target task. When this condition is weak, return relabeling may introduce additional bias, and incorporating relabeled source trajectories can degrade performance.
> > >
> > > This behavior is observed for **Reinformer** on **Walker2D Medium-Replay** under both **BodyMass** and **JointNoise** shifts, where REAG provides limited gains and, in some cases, leads to performance degradation. Medium-Replay datasets exhibit substantial **heterogeneity and multi-modality** due to mixtures of behavior policies and return distributions. In such settings, the target subset (1T) is often insufficient to reliably characterize the target return distribution, which can be **non-Gaussian** in parts of the Medium-Replay data. As a result, the estimated return statistics and value functions become noisy, reducing the effectiveness of return relabeling.
> > >
> > > Based on this analysis, we clarify in the revised manuscript that **REAG is best suited for settings with sufficient source–target occupancy overlap and relatively stable return distributions** (e.g., Medium and Medium-Expert datasets). For highly heterogeneous replay datasets, we recommend using $\text{REAG}^{\mathrm{Dara}}_{\mathrm{MV}}$ or disabling return augmentation. This failure-mode analysis has been included in **Section E** of the revised manuscript.

---

### Review · Reviewer_tdvw · 2025-12-11

**Summary Of Contributions:**

This paper addresses the dynamics shift problem in return-conditioned supervised learning (RCSL), highlighting challenges arising from the peculiar structure of the RCSL policy class and the lack of a straightforward policy representation. To mitigate these issues, the authors propose a return-augmented approach for Decision Transformers, in which the return signal in the source domain is modified to align its distribution with that of the target domain. They introduce two instantiations of this approach: REAG_Dara and REAG_MV. The authors further provide theoretical guarantees showing that, under this alignment, the learned policy achieves the same suboptimality bound as a policy trained directly on data from the target domain.

Note: I am not familiar with this line of research and cannot asses the novelty or positioning of the paper in literature.

**Audience:**

Yes

**Audience Explanation:**

This paper introduces a method to improve performance of decision transformers in off dynamics reinforcement learning, which is an interested topic in the community.

**Claims And Evidence:**

Yes

**Claims Explanation:**

This paper provides theoretical and empirical evidence to support the claims.

**Requested Changes:**

Minor issues : There is a typo in preliminary section - plolicy

---

> ### Author Response · Authors · 2026-01-06
> **Response to Reviewer tdvw**
>
> > **Q1: Minor Issues**
> > There is a typo in the Preliminary section (“plolicy”).
>
> **Response.**
> Thank you for pointing this out. We have corrected this typo in the revised manuscript.

---

### Review · Reviewer_bpTG · 2025-12-16

**Summary Of Contributions:**

This paper studies offline off-dynamics RL with abundant source-domain data and scarce target-domain data, extending return-conditioned supervised learning (RCSL), particularly Decision Transformer (DT), to leverage mismatched dynamics. It proposes return-augmented RCSL (REAG), with two instantiations: (i) REAG-DARA, which applies density-ratio–based per-step augmentation summed into returns, and (ii) REAG-MV, which aligns source and target conditional return distributions via moment matching using offline Q estimates. A finite-sample analysis shows that, under standard RCSL assumptions and domain overlap, incorporating augmented source data improves the bound over target-only learning.

Strengths
- Novel extension of off-dynamics augmentation to return-conditioned policies, where standard reward correction methods (DARC/DARA) do not directly apply.

- Clear return-distribution alignment perspective for RCSL, with practical instantiations connecting prior trajectory matching and DT-style learning.

Limitations:
- The REAG-MV theory assumes access to conditional return CDFs under the same behavior policy; in practice, moments are approximated via learned Q-functions and across-action variability, which weakens the connection to the theoretical formulation.

- The “same behavior policy” assumption is violated in the datasets used, so theoretical guarantees for the implemented estimator are unclear.

- Dropping DT’s return-consistency constraint improves performance but changes return-token semantics and may risk brittleness or extrapolation issues.

**Audience:**

Yes

**Audience Explanation:**

Real data rarely comes from the exact environment you care about. The paper shows how you can reuse lots of existing data collected under different dynamics instead of throwing it away. Both the theory and experiments suggest when mixing source and target data is beneficial, which is useful for researchers working with limited target data.

**Claims And Evidence:**

Yes

**Claims Explanation:**

Mostly yes. The empirical results are generally convincing and consistently show that the proposed return-augmentation methods improve DT-type baselines across multiple off-dynamics benchmarks and ablations. However, for REAG-MV in particular, the theoretical claims rely on strong assumptions (shared behavior policy, access to conditional return distributions) that are violated in practice, and the implemented estimator only loosely corresponds to the theory.

**Requested Changes:**

Methodological issues
- Missing comparisons to several strong and recent off-dynamics baselines tailored to large dynamics gaps: model-based/planner approaches (e.g., OCDP), target-dynamics modeling (MOBODY), robust cross-domain Q-learning (DROCO), and sequence-model cross-domain pipelines (DFDT). Including at least one of these would situate gains more convincingly.

- Sensitivity to the quality of the learned Q used in REAG*MV is only lightly probed; CQL trained on tiny target sets can be unstable. More systematic sensitivity analyses (vary target size, critic seeds, α values) would strengthen claims.

Presentation Issues
- Notation occasionally shifts (g vs g(tau) vs g(s_t) Ppi vs ppi) and equations include minor typesetting issues that obscure exact meanings (4.6)

- Some tables contain formatting artifacts (arrows embedded in text, inconsistent symbols) and are hard to parse; figures lack axis units in places and have inconsistent labeling (“REAGMY” vs “REAGMV”).

Missing related work or comparisons
- There appears to be substantial overlap with prior work on Return Augmented Decision Transformer (RADT), which proposes conceptually similar RADT‑DARA and RADT‑MV variants and a closely related theoretical result. This paper does not cite or differentiate from that work; clarifying novelty, differences in theory/algorithms/experiments, and empirical deltas is necessary.

---

> ### Author Response · Authors · 2026-01-06
> **Response to Reviewer bpTG (1/2)**
>
> >**Q1: Theory–Practice Gap**
> > 1. The REAG-MV theory assumes access to conditional return CDFs under the *same* behavior policy; in practice, moments are approximated via learned Q-functions and across-action variability, which weakens the connection to the theoretical formulation.
> > 2. The “same behavior policy” assumption is violated in the datasets used, so theoretical guarantees for the implemented estimator are unclear.
>
> **Response.**
> Thank you for this insightful question. We acknowledge the gap between the theoretical formulation and the empirical implementation.
> Our theoretical analysis is intended to provide *principled motivation and conceptual guidance* for the proposed method. In practice, exact access to conditional return CDFs is unavailable, and the same-behavior-policy assumption is not strictly satisfied in the datasets we consider. As a result, we approximate the relevant quantities using learned Q-functions and across-action variability. This approximation improves numerical stability and empirical performance, but it weakens the direct correspondence to the idealized theoretical setting.
> To better align theory and practice, in the revised manuscript (**Appendix F**), we generalize our analysis to the setting where the source and target datasets are generated by *different* behavior policies. This extension more closely matches the empirical implementation and clarifies the theoretical interpretation of the proposed estimator under realistic data collection conditions.
>
> > **Q2: Dropping DT’s Return-Consistency Constraint**
> > Dropping DT’s return-consistency constraint improves performance but changes return-token semantics and may risk brittleness or extrapolation issues.
>
> **Response.**
> Thank you for the comment. We are not the first to relax DT’s return-consistency constraint. Prior work, such as **Reinformer** [1] and **Reinforced RCSL** [2], has shown that dropping this constraint can improve trajectory stitching and lead to better empirical performance.
>
> In our setting, we explicitly evaluate this design choice through an ablation study (Figure 5). The results show that enforcing return consistency does not improve performance and, in most cases, leads to worse outcomes. This suggests that relaxing the constraint does not introduce brittleness in the off-dynamics setting considered.
>
> **References**
> - [1] Zhuang, Z., et al. *Reinformer: Max-return sequence modeling for offline RL.* arXiv:2405.08740, 2024.
> - [2] Liu, Z., et al. *How to Provably Improve Return Conditioned Supervised Learning?* arXiv:2506.08463, 2025.
>
>
> > **Q3: Missing Comparisons to Recent Off-Dynamics Baselines**
> > Missing comparisons to several strong and recent off-dynamics baselines tailored to large dynamics gaps: model-based/planner approaches (e.g., OCDP), target-dynamics modeling (MOBODY), robust cross-domain Q-learning (DROCO), and sequence-model cross-domain pipelines (DFDT). Including at least one of these would situate gains more convincingly.
>
> **Response.**
> We include **DFDT** as an additional baseline in our evaluation. DFDT also builds upon a Decision Transformer–based architecture to address off-dynamics (cross-domain) reinforcement learning, but it differs from our approach in its core design. While our method emphasizes **return augmentation**, DFDT focuses on **source data selection** via a two-level, transition-level filtering strategy.
>
> All DFDT results are reported under the same experimental settings as our method to ensure a fair comparison. Specifically, we evaluate DFDT on the **Walker2D**, **Hopper**, and **HalfCheetah** environments under the **medium**, **medium-replay**, and **medium-expert** datasets. In our setup, the source environments are constructed by introducing **body-mass** and **joint-noise** perturbations, while the target environment retains the original, unshifted dynamics.
>
> Quantitative results are summarized in **Table 9** of the revised manuscript, and an overall comparison with other methods is presented in **Figure 1**. Across these benchmarks, our method, $\text{REAG}_{\mathrm{MV}}^{\mathrm{QT}}$, consistently outperforms DFDT. These results demonstrate the effectiveness of our approach and its competitive performance relative to recently proposed off-dynamics reinforcement learning methods.

---

> > ### Author Response · Authors · 2026-01-06
> > **Response to Reviewer bpTG 2/2**
> >
> > > **Q4: Sensitivity to Q-Function Quality**
> > > Sensitivity to the quality of the learned Q used in REAG*MV is only lightly probed; CQL trained on tiny target sets can be unstable. More systematic sensitivity analyses (vary target size, critic seeds, α values) would strengthen claims.
> >
> > **Response.**
> > Thank you for the suggestion. We have conducted additional systematic ablation studies to examine the sensitivity of **REAG*MV** to the quality of the learned Q-functions used for return relabeling. Specifically, we vary (i) the conservative regularization parameter $\alpha$ in CQL and (ii) the number of samples used to train the CQL model.
> >
> > These experiments are performed on **Walker2D Medium** under the **BodyMass** dynamics shift setting. The results, presented in **Figure 7** of the revised manuscript, show that $\text{REAG}_{\mathrm{MV}}^{*}$ exhibits stable performance across a wide range of $\alpha$ values and training sample sizes. Despite variability in the learned CQL policies, the downstream policy performance remains largely consistent.
> >
> > Overall, these results indicate that **REAG*MV** is not highly sensitive to instability in CQL training, supporting the robustness of the proposed approach.
> >
> > > **Q5: Notation, Typesetting, and Presentation Issues**
> > > Notation occasionally shifts (e.g., $g$ vs. $g(\tau)$ vs. $g(s_t)$, $P_\pi$ vs. $p_\pi$), and some equations contain minor typesetting issues that obscure exact meanings (e.g., Eq. 4.6). Some tables contain formatting artifacts (e.g., arrows embedded in text, inconsistent symbols) that are hard to parse; figures lack axis units in places and have inconsistent labeling (e.g., “REAGMY” vs. “REAGMV”).
> >
> > **Response.**
> > Thank you for the careful review and detailed feedback.
> >
> > - **Minor typesetting issues (Eq. 4.6).**
> > We have either revised the notions or added necessary definitions to ensure that all symbols and expressions are clearly defined and unambiguous.
> >
> > - **Table formatting artifacts.**
> >   The arrows in the tables are intentional and convey specific meanings: performance changes due to augmentation are indicated with **red upward arrows** for improvements and **green downward arrows** for degradations. To improve clarity, we have explicitly explained this notation in the corresponding table captions and ensured consistent symbol usage throughout.
> >
> > - **Missing axis units in figures.**
> >   We have revised the figures to include the missing axis units and verified that all axes are clearly labeled.
> >
> > - **Inconsistent labeling (“REAGMY” vs. “REAGMV”).**
> >   We carefully checked the manuscript but did not find this specific typo. We would greatly appreciate it if the reviewer could point us to the exact location, and we will correct it immediately.

---

### Decision · Action_Editor_qyk9 · 2026-01-30

**Recommendation:** Accept with minor revision

**Additional Comments:**

Reviewers raised concerns mainly regarding the novelty and significance of this work. Although novelty is not a primary acceptance criterion for TMLR, it would still benefit the paper to more clearly articulate its contributions relative to prior work and to better highlight what is new in this setting. Besides, reviewers noted several experimental issues, including the limited experimental scope and the lack of baselines.

While these concerns do not undermine the technical soundness of the paper, addressing them would improve clarity and strengthen its empirical support.

**Audience:**

Yes

**Audience Explanation:**

Though this paper does not introduce fundamentally novel techniques, it addresses a relevant problem in reinforcement learning, particularly in off-dynamics and return-conditioned policy learning, which is likely to be of interest to a subset of the TMLR audience working on offline RL.

**Claims And Evidence:**

Yes

**Claims Explanation:**

The claims are generally supported by theoretical analysis and empirical results. While some reviewers raised concerns regarding novelty and positioning, the technical arguments are sound, and the experimental results are consistent with the stated claims. Overall, the evidence is convincing under TMLR’s evaluation criteria.